# WRF-Comfort: Simulating micro-scale variability of outdoor heat stress at the city scale with a mesoscale model

Alberto Martilli[1], Negin Nazarian[2,3], E. Scott Krayenhoff[4], Jacob Lachapelle[4], Jiachen Lu[2,3], Esther Rivas[1], Alejandro Rodriguez-Sanchez[1], Beatriz Sanchez[1], José Luis Santiago[1]

[1]Atmsopheric Modelling Unit, Environmental Department, CIEMAT, Madrid, 28040, Spain
[2]School of Built Environment, University of New South Wales, Sydney, Australia
[3]ARC Centre of Excellence for Climate Extremes, Australia
[4]School of Environmental Sciences, University of Guelph, Guelph, Canada

*Correspondence to*: Alberto Martilli (alberto.martilli@ciemat.es)

**Abstract.** Urban overheating, and its ongoing exacerbation due to global warming and urban development, leads to increased exposure to urban heat and increased thermal discomfort and heat stress. To quantify thermal stress, specific indices have been proposed that depend on air temperature, mean radiant temperature (MRT), wind speed, and relative humidity. While temperature and humidity vary on scales of hundreds of meters, MRT and wind speed are strongly affected by individual buildings and trees, and vary at the meter scale. Therefore, most numerical thermal comfort studies apply micro-scale models to limited spatial domains (commonly representing urban neighborhoods with building blocks) with resolutions on the order of 1 m and a few hours of simulation. This prevents the analysis of the impact of city-scale adaptation/mitigation strategies on thermal stress and comfort. To solve this problem, we develop a methodology to estimate thermal stress indicators and their subgrid variability in mesoscale models - here applied to the multilayer urban canopy parametrization BEP-BEM within the WRF model. The new scheme (consisting of three main steps) can readily assess intra-neighborhood scale heat stress distributions across whole cities and for time scales of minutes to years. The first key component of the approach is the estimation of MRT in several locations within streets for different street orientations. Second, mean wind speed, and its subgrid variability, are downscaled as a function of the local urban morphology based on relations derived from a set of microscale LES and RANS simulations across a wide range of realistic and idealized urban morphologies. Lastly, we compute the distributions of two thermal stress indices for each grid square combining all the subgrid values of MRT, wind speed, air temperature, and absolute humidity. From these distributions, we quantify the high and low tails of the heat stress distribution in each grid square across the city, representing the thermal diversity experienced in street canyons. In this contribution, we present the core methodology as well as simulation results for Madrid (Spain), which illustrate strong differences between heat stress indices and common heat metrics like air or surface temperature, both across the city and over the diurnal cycle.

## 1 Introduction

The combination of urban development and climate change has increased heat exposure in cities in recent decades (Tuholske et al., 2021) and a continuation of these trends in the 21st century would be difficult to offset locally from an air temperature perspective (Broadbent et al., 2020; Krayenhoff et al., 2018; Zhao et al., 2021). Adaptation options that target contributions to heat exposure other than the air temperature, such as radiation (e.g., via shade) and wind (e.g. via improved street ventilation), should therefore be considered. Quantification of these contributions relative to air temperature requires the application of comprehensive thermo-physiological heat stress metrics such as the Universal Thermal Climate Index, UTCI (Jendritzky et al., 2012), the Physiological Equivalent Temperature, PET (Höppe, 1999), or the Standard Effective Temperature, SET (Gagge et al., 1986). Moreover, exposure to heat hazards is moderated by infrastructure-based and social/mobility-based adaptations to heat, and by buildings and associated cooling mechanisms. Here, the focus is the development of a tool to quantify the outdoor component of heat exposure in cities, accounting for all relevant meteorological variables.

Heat exposure in urban areas is affected by several meteorological variables that vary on different spatial and temporal scales (Nazarian et al., 2022). While temperature and humidity vary on spatial scales on the order of hundreds of meters, shortwave and longwave radiation and wind speed are strongly affected by individual buildings and vary at the scale of a few meters. For this reason, most numerical thermal comfort studies in urban areas apply micro-scale models with resolutions on the order of one m and spatial domains that are limited to an urban block or neighborhood (Nazarian et al., 2017; Zhang et al., 2022; Geletič et al., 2018). While these studies include substantial detail at the micro-scale, they are very expensive computationally and therefore can be applied only to a few neighborhoods and they neglect the interactions with larger scale meteorological phenomena (e.g., land/sea breezes, mountain/valley winds, urban breezes) that often play a relevant role in outdoor thermal comfort and its variation across cities. On the other hand, contemporary meso-scale numerical models can be applied to the whole urban area and surrounding regions, and therefore capture these larger-scale phenomena, but have spatial resolutions of several hundred meters at best. These models use a grid mesh that does not resolve buildings and is therefore too coarse to capture the fine-scale variation of radiation and wind flow of relevance to outdoor heat exposure and ultimately thermal comfort.

The objective of this work is to fill the aforementioned gap by developing a model that includes the most crucial capabilities of micro-scale assessments of thermal exposure within meso-scale models. This new model will quantify the spatial variability (i.e., statistical representation of the microscale distribution) for longwave and shortwave radiation as well as wind speed within each meso-scale grid square. Subsequently, it will capture the range of thermal exposure, as quantified by the UTCI and SET thermal stress metrics, within each urban grid square across a city at each time of day. The focus here is on the *range* of thermal exposure, such that we identify the cool and hot spots within the grid cell without having to resolve the entire spatial distribution. We argue that this represents the most crucial information for heat management and urban design interventions, as it identifies whether people can move and search for optimal thermal conditions. For example, if hot spots are experiencing extreme heat stress but the cool spots are at slight heat stress, pedestrians have the opportunity, and autonomy, to seek shade

and thermal respite (i.e., temporal and spatial autonomy as described in Nazarian et al. (2019)). Conversely, if the conditions in the cool spot are already in extreme heat stress, this can be used to inform urban design interventions or heat advisories to vulnerable populations to avoid being outside at that place and time. Overall, representing the range of heat exposure at the neighborhood scale while covering regional-scale phenomena is key to human-centric assessments of urban overheating (Nazarian et al., 2022).

The new model is embedded in the multi-layer urban canopy parameterization BEP-BEM (Martilli et al., 2002; Salamanca et al., 2010) which simulates the local-scale meteorological effects of the grid-average urban morphology within the Weather Research and Forecasting (WRF) mesoscale model (Skamarock et al., 2019 version 4.3 has been used in this study). Here, BEP-BEM is extended to quantify the spatial variation of the mean radiant temperature and wind speed within the grid square at the pedestrian level. To our knowledge, three schemes in the published literature have attempted to capture thermal exposure in an urban canopy model. Pigliautile (2020) implemented a scheme to estimate human thermal exposure in the Princeton Single-Layer Urban Canopy Model. However, the scheme has not been run within a mesoscale model. Jin et al. (2022) calculate urban mean radiant temperature (MRT) in a mesoscale model, while Lemonsu (2015) and Leroyer et al. (2018) assess UTCI in mesoscale modeling applications within Paris and Toronto, respectively. Moreover, Giannaros et al (2018, 2023), made an offline coupling of WRF-BEP_BEM with RayMan (Matzarakis et al. 2007). However, none of these approaches account for the within-grid spatial variation of wind speed, and their assessment of sub-grid spatial variation of radiation exposure (i.e., mean radiant temperature) is limited. Here, we further extend the BEP-BEM model embedded in the WRF meso-scale model to overcome these limitations and better assess spatial variation of thermal exposure within each urban grid square.

In section 2, the methodology is described in detail, with a focus on model development and implementation in WRF. In Section 3, we present an example of the type of outputs that can be produced. Conclusions are in section 4.

## 2 Methodology

The most complete thermal stress indices invariably depend on four meteorological variables: air temperature, mean radiant temperature (MRT), relative humidity, and wind speed. Among these, MRT and wind speed have the largest spatial variability in the urban canopy, and this variability is often captured with 3D micro-scale models of urban airflow and radiative heat transfer. At the meso-scale, however, it is not feasible to incorporate such models, motivating the simplified urban canopy parameterizations developed here. Below we detail how the BEP-BEM urban canopy model is modified to a) introduce a simplified model for MRT variation within a meso-scale grid cell (Sec. 2.1) and b) parameterize airflow variability (Sec. 2.2) in the urban canopy within a grid cell, and make a simple estimate of air temperature variability. These meteorological parameters are then used to estimate the sub-grid scale variation of thermal stress indices (Sec. 2.3), namely SET and UTCI, as two of the most commonly used indices for outdoor environments (Potchter et al 2018). Lastly, we discuss how multi-scale

temporal and spatial variabilities in thermal exposure can be effectively communicated using the outcomes of the updated
WRF-BEP-BEM model.

## 2.1    A simplified model for MRT variability in the urban canopy

The mean radiant temperature is a measure of the total radiation flux absorbed by the human body, including both shortwave
(from the sun, either directly or after reflection on the walls or road) and longwave (emitted from solid bodies like walls or
road, or from the sky) radiation. Whether pedestrians are shaded or in the sunshine, as well as their distance from warm surfaces
emitting radiation, is therefore very important. BEP-BEM applies a simple urban morphology: two street canyons of different
orientations, each with the same street width and building height distribution on each side of the canyon (Martilli et al. 2002).
To capture the within-grid spatial extremes of mean radiant temperature, we assess pedestrian locations at the center of the
street for two canyon orientations considered in BEP-BEM and at positions located at a distance of 1.5 m from the building
wall on each side of the street, representing the sidewalks. Thus, there are 6 positions (three for each street direction) in each
urban grid square where we compute the mean radiant temperature (shown for the example of North-South and East-West
streets in Fig. 1). For shortwave and longwave radiation exchange, the standard BEP view factor and shading routines (Martilli
et al. 2002) are used to estimate the amount of shortwave (direct and diffuse) and longwave radiation reaching a vertical
segment 1.80 m tall and located in each of the six positions previously mentioned (Fig. 1, Appendix A). Reflection of shortwave
radiation and emission and reflection of longwave radiation from both building walls and the street surface are accounted for
via these view factors. The pedestrian is "transparent" from the perspective of the urban facets, meaning that its presence does
not alter the shortwave and longwave radiation reaching the building walls and road. The mean radiant temperature is computed
by weighting the radiation reaching each side of the vertical segment by 0.44, and the radiation reaching the downward- and
upward-facing (at 1.80 m height) surfaces of the pedestrian by 0.06 each. This approach follows the six-directional weighting
method (Thorsson et al. 2007) and aggregates the four lateral weightings of 0.22 into two lateral weightings of 0.44 since BEP-
BEM is a two-dimensional model (e. g. the streets are considered infinitely long). Namely,

$$T_{MRT} = \sqrt[4]{\frac{\sum_{i=1,4} W_i(a_K K_i + a_L L_i)}{a_L \sigma}}$$

115    (1)

where, for a N-S oriented street, i=1,2 are for the vertical sides of the pedestrian looking East, and West respectively, and i=3,4
are for the top and bottom. Therefore, $W_{1,2}$=0.44, while $W_{3,4}$=0.06, while the absorptivity of the pedestrian in the shortwave
and longwave, $a_K$ (the absorption coefficient for shortwave radiation of the human body) and $a_L$ (the absorption coefficient for
long-wave radiation, or emissivity, of the human body), respectively, are $a_K$=0.7, and $a_L$=0.97, $K_{1,2}$ and $L_{1,2}$ are the short and
longwave radiation reaching the vertical segment, and $K_{3,4}$ and $L_{3,4}$ are short and longwave radiation reaching the top and
bottom respectively, and σ is the Stefan-Boltzmann constant (see Appendix A for details about how the radiation components
are computed).

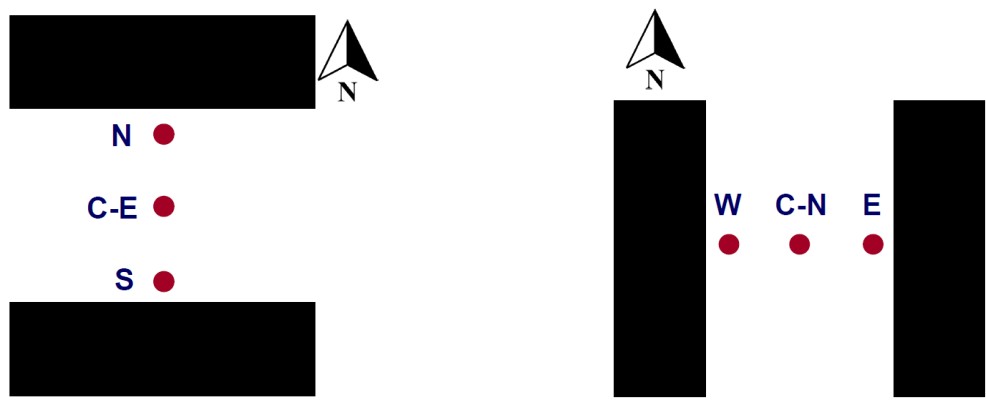

**Figure 1:** Two street directions (left: E-W canyon, right: N-S canyon) and pedestrian locations considered for Mean Radiant Temperature calculations.

The diurnal progression of the mean radiant temperature computed by this new model in BEP-BEM is subsequently compared
with that obtained from TUF-Pedestrian, a more detailed three-dimensional model that has been evaluated against
measurements (Lachapelle et al. 2022; Jiang et al. 2023). TUF-Pedestrian is configured with identical input parameters and
meteorological forcing, and with long canyons that approximate the two-dimensional BEP-BEM canyon geometry. The new
model clearly captures the relevant details of the diurnal progression of MRT at all six locations (Fig. 2), with a mean absolute
difference of 3.4 K, and a root mean square difference of 4.3 K across all locations. A comparison of the shortwave radiation
loading on the pedestrian between the two models reveals very good agreement (Appendix B Fig. B1, B2), considering the
highly simplified urban morphology used by BEP-BEM, with biggest errors limited to short periods of time; thus, most of the
model disagreement arises from differences between longwave loading on the pedestrian as a result of different methods for
computation of surface temperature between the models. Overall, the new model of mean radiation temperature in BEP-BEM
provides satisfactory results.

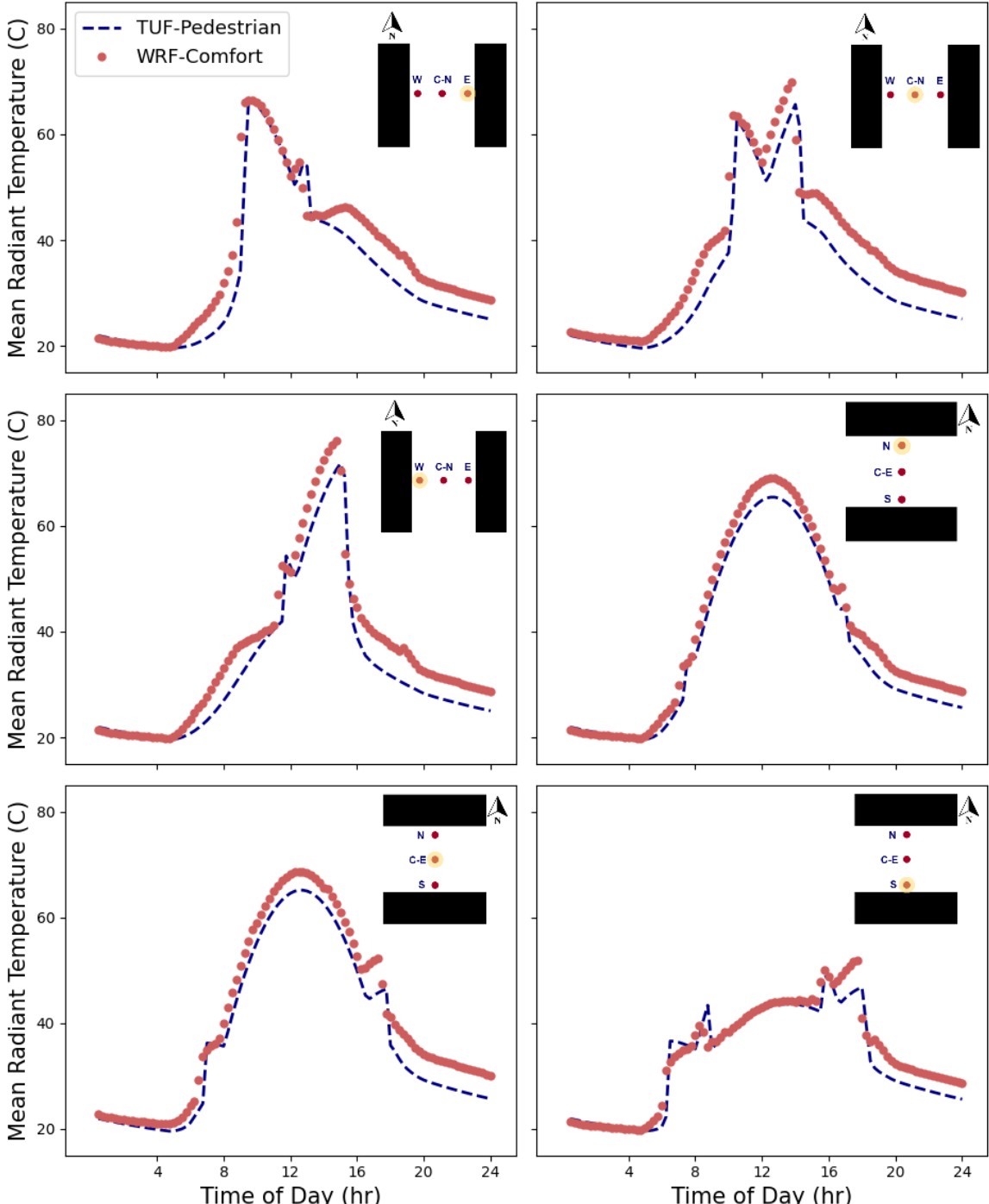

**Figure 2:** Comparison of diurnal variation of Mean Radiant Temperature (MRT) between the new model in BEP-BEM and TUF-Pedestrian for each of the six locations in Fig. 1. TUF pedestrian acts here as a reference.

## 2.2 Parameterize airflow variability in the urban canopy

Mesoscale models solve conservation equations for the three components of momentum. From these, it is possible to derive the spatially averaged wind velocity in each grid cell, at the grid resolution of the mesoscale model, commonly of the order of 300m-1km. The spatially averaged wind velocity in the urban canopy $\langle V \rangle$, close to the pedestrian height (~2.5m), is the square root of the sum of the spatial average of the two horizontal components $u$, and $v$, (neglecting the vertical component, which is usually at least one or two orders of magnitude smaller than the horizontal),

$$\langle V \rangle = \frac{1}{V_{air}} \sqrt{\left( \int_{V_{air}} u dV \right)^2 + \left( \int_{V_{air}} v dV \right)^2} \tag{2}$$

where here $V_{air}$ is the volume of the grid cell occupied by air (e. g. without the buildings)

However, the wind velocity calculated in mesoscale models is different from the average wind speed that would be experienced by a person in the grid cell. This is better represented by the spatial average of the wind speed $\langle U \rangle$ (e. g. the modulus of the vector), written as

$$\langle U \rangle = \frac{1}{V_{air}} \int_{V_{air}} \sqrt{u^2 + v^2} dV \tag{3}$$

To assess the impact of airflow on human thermal comfort, the wind speed should be estimated from the wind velocity computed in the mesoscale models. Additionally, it is critical to parameterize and estimate the spatial variability of mean wind speed in the urban canopy. Accounting for these factors, the range of wind speed variability at the pedestrian level is estimated, which is critical for the quantification of spatial variability of outdoor thermal stress and comfort.

Here, we describe the parameterization of a) wind speed-to-velocity ratio and b) wind speed distribution, based on urban density parameters. Data are considered from over 173 microscale CFD simulations of urban airflow over realistic and idealized urban configurations, spanning a wide range of building plan area ($\lambda_P$), frontal area ($\lambda_F$), and wall area ($\lambda_w$) densities representative of realistic urban neighbourhoods in different types of cities. CFD simulations are conducted using 162 large-eddy simulations (LES) and 11 Reynolds-averaged Navier–Stokes (RANS) schemes detailed in Appendix C.

Mean wind velocity $\langle V \rangle$, speed $\langle U \rangle$ and its spatial standard deviation ($\sigma_U$) are computed at a horizontal cross-section at pedestrian height for each CFD simulation and used for deriving parameterizations (Fig. 3). An additional data point is added at $\lambda_P = \lambda_w = 0$, ensuring that wind speed is equal to wind velocity, and its standard deviation is set to zero, for the non-urban case. It is important to remark here, that we are dealing with the standard deviation of the spatial distribution of the mean wind speed. With the term *mean* we indicate the result of an ensemble (over many realizations) or time average (over time scales larger than the turbulence time scale, but smaller than the time scale of the mesoscale motions), but not a spatial average. The urban canopy in fact is spatially heterogeneous, and, for this reason, the time and ensemble averages are different than the spatial average. Only when $\lambda_P = \lambda_w = 0$ (e. g. there are no buildings), and the horizontal homogeneity is recovered, must the

variability be zero. This $\sigma_U$, therefore, should not be confounded with the turbulent $\sigma$, which indicates the variability in
instantaneous wind speed induced by turbulent motions, which indeed is not zero even when there are no buildings.

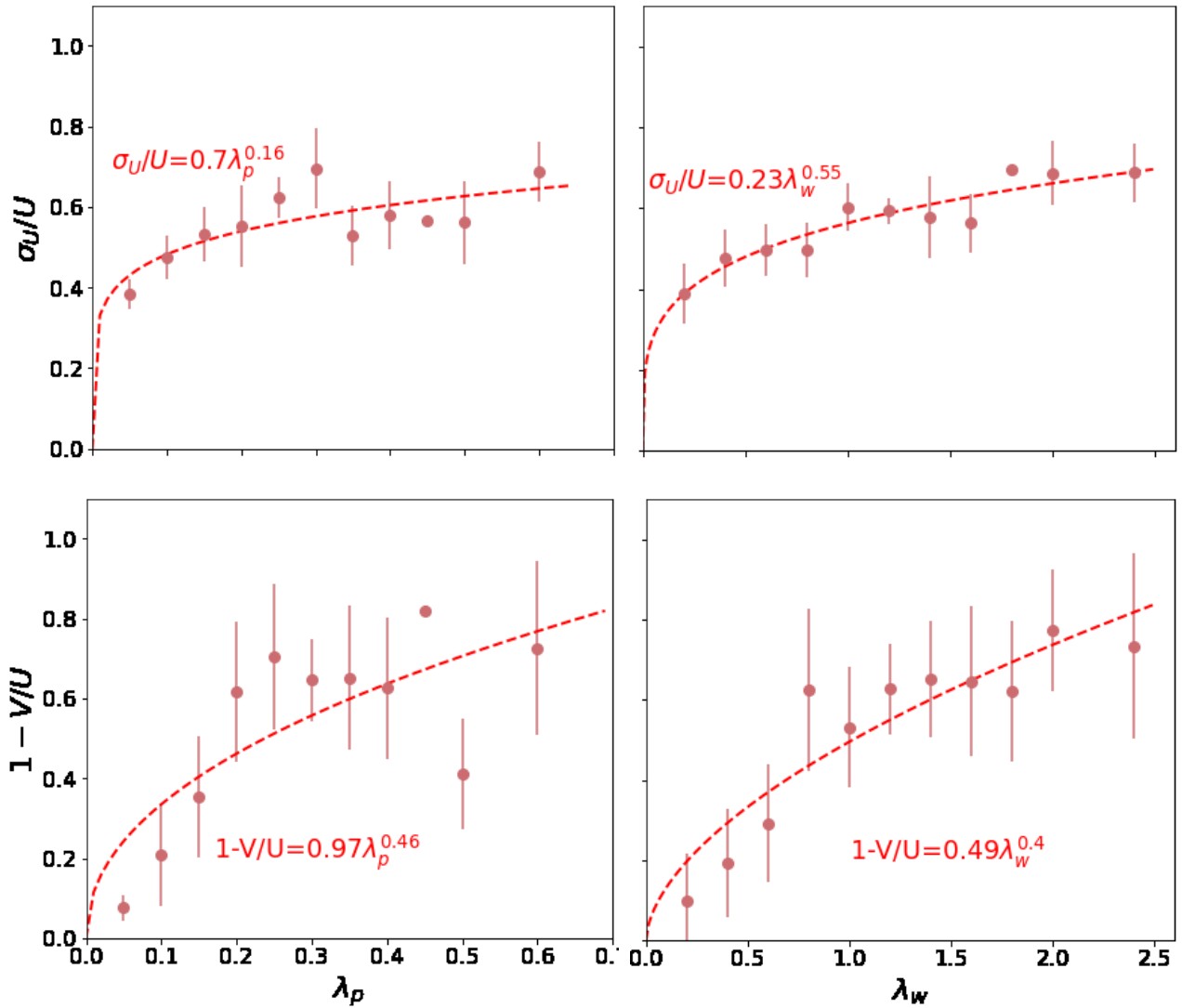

**Figure 3:** Relationship between 1-<V>/<U> (bottom row), and $\sigma_U$/<U> (top row), and two morphological parameters, $\lambda_P$ (left column), and
$\lambda_W$ (right column) based on the CFD simulations. Dots represent the average of the value among all the simulations that share the same
morphological parameter, and the vertical bar indicates the standard deviation. The dashed line and the formula indicate the best fit.

Parameterizations are derived (shown in Fig. 3) for two density parameters ($\lambda_P$=Ap/Atot, and $\lambda_w$=Aw/Atot, where Ap is the
area of the horizontal surface occupied by buildings, or the roof area, Aw is the area of vertical (wall) surfaces, and Atot is the
total horizontal area). We find that $\lambda_w$ better predicts mean wind speed and its spatial variability at the pedestrian height,
because it represents both horizontal and vertical heterogeneities in the urban canopy. Note that $\lambda_F$ has not been included in
the study, given the difficulty to estimate it for real urban areas, and to translate it to the simplified 2D urban morphology used
by BEP-BEM. In any case, $\lambda_F$ is closely related to $\lambda_w$. Therefore, the following parameterizations are implemented at the
pedestrian height as a function of the wall area density $\lambda_w$

$$\langle U \rangle = \frac{\langle V \rangle}{1 - 0.49\lambda_w^{0.4}} \tag{4}$$
$$\sigma_U = \langle U \rangle (0.25\lambda_w^{0.55}) \tag{5}$$
We, therefore, assign three values of wind speed in each grid cell,
$$\langle speed \rangle_1 = max(0.01, \langle U \rangle (1 - 0.25\lambda_w^{0.55}))$$
$$\langle speed \rangle_2 = \langle U \rangle \tag{6}$$
$$\langle speed \rangle_3 = \langle U \rangle (1 + 0.25\lambda_w^{0.55})$$
Note that here we consider the three values equally likely, in order to realistically span the range of possible values that the
wind speed can take in each grid cell. Since UTCI has been designed for 10m wind speeds, a simple log law is used to
rescale wind speed at 10m, before passing it to the UTCI routine.
**2.3    Calculation of the thermal comfort index**
To represent the subgrid spatial variability of air temperature, detailed CFD simulations are not available, so we simply use a
variability of 1 degree Celsius, which we consider to be a conservative estimate of the spatial variability of air temperature
over a spatial scale of the order of one km$^2$This value is consistent with the range obtained in the few non-netural simulations
available, like Santiago et al. (2014), and Nazarian et al. (2018) over idealized arrays, as well as that obtained by Rivas-
Ramos over a realistic neighbourhood of Madrid (2024, personal communication). A better determination of the variability is
left to future studies. Therefore, for each grid cell, we have three values for air temperature:
$$Temp_1 = Temp_{WRF} - 1$$
$$Temp_2 = Temp_{WRF} \tag{7}$$
$$Temp_3 = Temp_{WRF} + 1$$
where $Temp_{WRF}$ is the air temperature provided by WRF.

We therefore have, for each urban grid cell, *three* values of wind speed, *three* values of temperature, and *six* values of mean radiant temperature. No variability of the absolute humidity is considered, but the relative humidity is computed using the three values of air temperature.

Based on the variation of these climate variables, assumed uncorrelated, 54 possible combinations of the air temperature, mean radiant temperature, and wind speed values can be formed. For each one of these combinations, we calculate the corresponding SET or UTCI value. Based on the resulting distribution, we estimate the value of the 10th, 50th, and 90th percentile SET or UTCI for each grid square (at each output time). Increasing the number of points where the mean radiant temperature is computed, or adding more values for the wind speed, does not change significantly the values of the percentiles (not shown).

## 3.    Characterization of thermal comfort in regional-scale models: Madrid case

To illustrate the capabilities of the new scheme, a typical heat wave day in the city of Madrid (Spain) is simulated with WRF. Madrid is located on a plateau at 500-700m above sea level, in the middle of the Iberian Peninsula. It experiences hot summers, with frequent heat waves that increasing cause severe heat stress in the population, and it is therefore considered a relevant case study. Four nested domains have been used, with resolutions of 27, 9, 3, and 1km respectively. The city morphology (Fig. 4) is derived from high-resolution LIDAR data that covers most of the metropolitan area of Madrid (Martilli et al., 2022), while the morphology of the surrounding towns is determined based on Local Climate Zone maps (Brousse et al., 2016). It is also important to mention that the city is located on a hilly terrain, with higher elevations in the N-W part of the urban area (around 700m a.s.l.) dropping to 500m a.s.l. or less in the S-E. Moreover, there are two topographical depressions on the two sides of the city centre, caused by the rivers Jarama and Manzanares (for a detailed description of the topography see also Martilli et al. 2022, where the same set-up was used). Other model configurations are the NOAH vegetation model for the non-urban grid points and the Bougeault and Lacarrere (1989) PBL scheme for turbulence parameterization. WRF coupled with BEP-BEM has previously been successfully used to simulate a heat wave period in Madrid (Salamanca et al., 2012). The period used in this paper is three days (14-16 July 2015). In particular, the analysis will focus on the 15th, when the maximum simulated temperature was above 40 Celsius. More information about the validation and a sensitivity study to select the optimal set-up can be found in Rodriguez-Sanchez (2020).

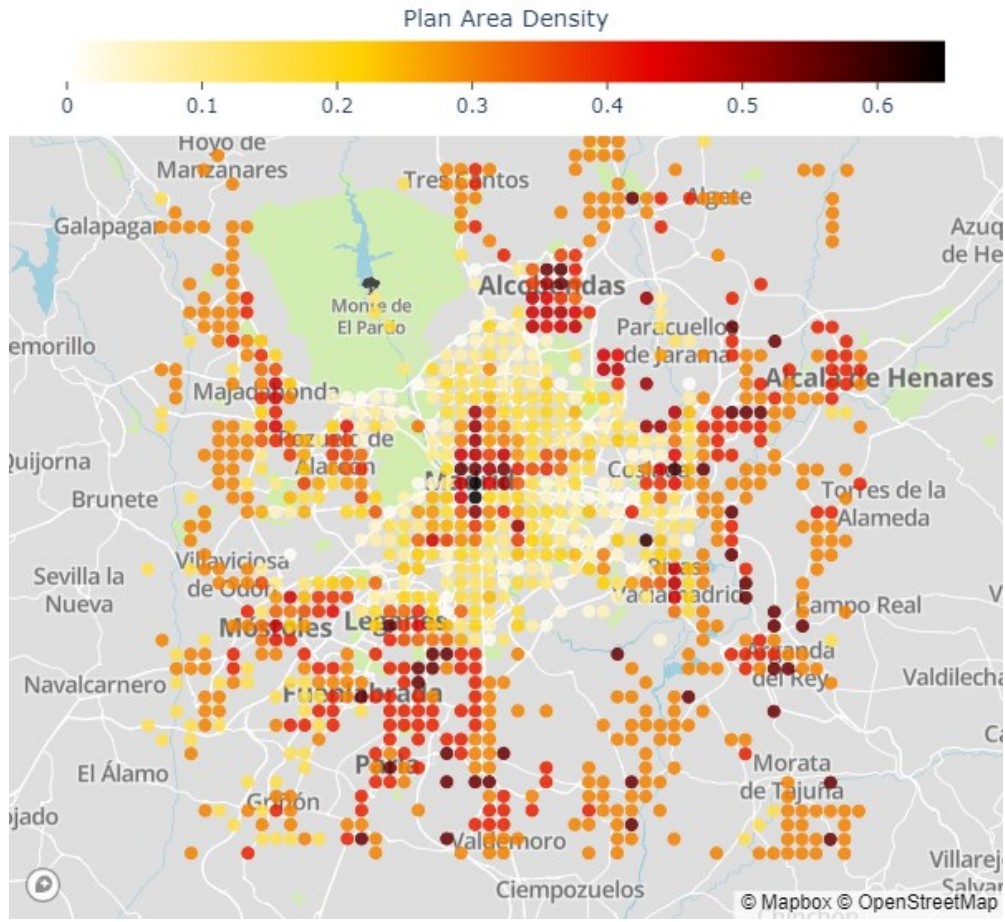

226

**Figure 4.** Map of the plan area building density over the Madrid region. The underlying map was created with Mapbox OpenStreetMap. The map is oriented such that left is west, and up is north; the size is 50x50km.

### 3.1 Sub-grid scale variability of MRT and thermal comfort.

In order to understand how urban morphology affects the simulated heat stress, we focus on two grid points with very different urban morphology. One is located in the dense core of the city, with a building plan area density of $\lambda_P$ =0.69, and a height-to-width ratio (H/W) value of 1.6. The second is located in the southern part of the urban area, in a residential neighbourhood with a much lower building density ($\lambda_P$ =0.2) and a H/W=0.1.

In Figure 5, the diurnal evolution of the mean radiant temperature in the six points (three per street direction) is presented for the high urban density point and the low urban density point. During the daytime, the impact of the shadowing is clear, with reduced mean radiant temperature in the high-density point compared to the more exposed low-density. On the other hand, during night-time, the reduced sky-view factor in the high-density point slows down the cooling compared to the more open low-density location.

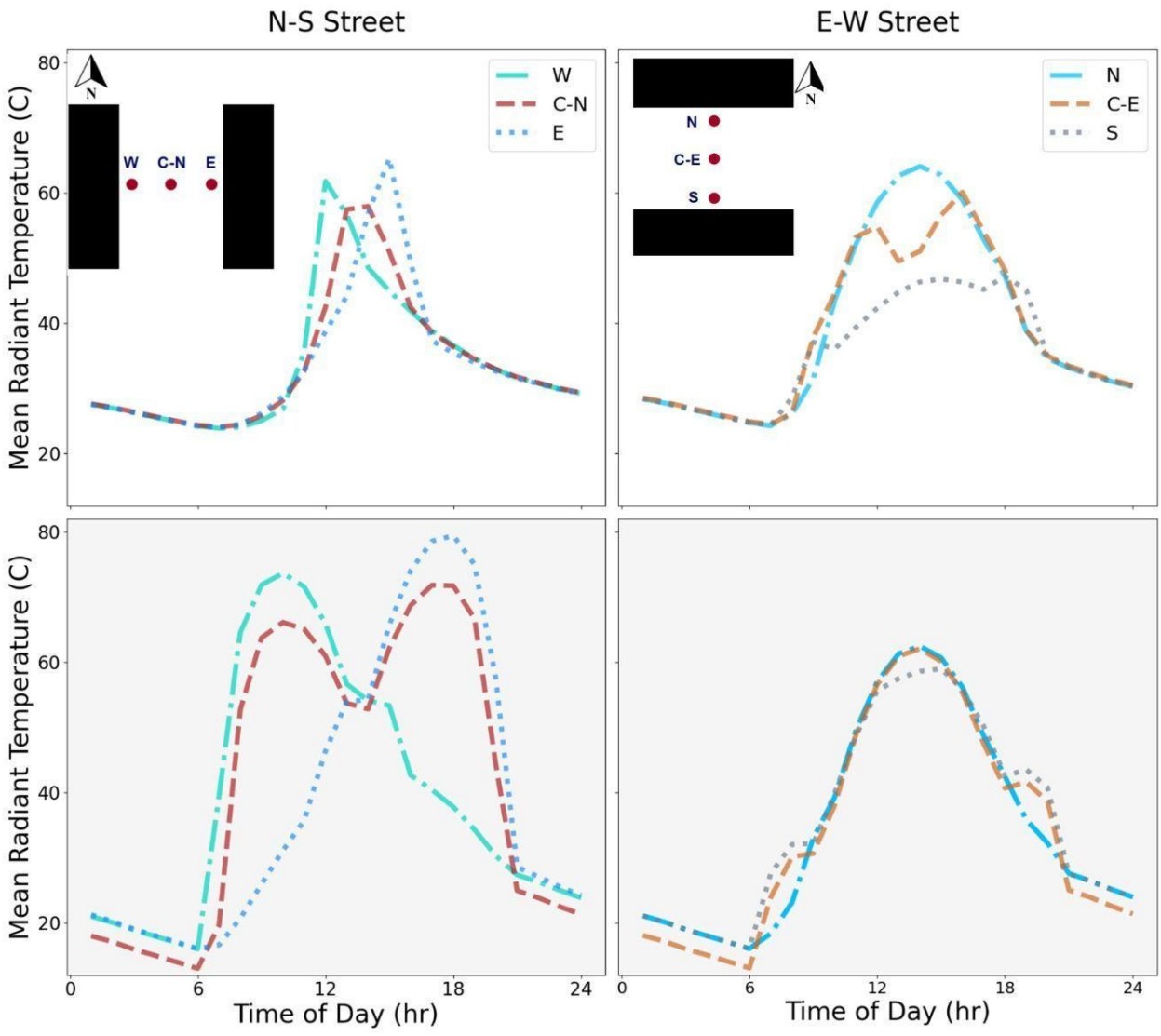

**Figure 5.** Diurnal evolution of MRT for 6 points in the urban canopy. The top row (white background) corresponds to a grid point with the highest building density in the centre of Madrid ($\lambda_P$ =0.69) while the bottom row (with grey background) shows MRT in a low-density neighbourhood ($\lambda_P$ =0.19). The left column is for an N-S street, while the right column shows an E-W street.

This behaviour helps to explain the heat stress index (Figure 6), which is introduced here as an example of an index that can be computed with standard outputs from meteorological models, i.e., without information related to the radiation environment

(e.g., MRT) and urban morphology. The air temperature indicates hotter values both during the day and the night in the high
urban density point compared to the low-density location. The Heat Index, which considers air temperature and humidity only,
and does not include mean radiant temperature or wind, shows the same tendency. On the other hand, the UTCI behaviour
communicates a different and more complete result. In the low-density neighbourhood, more exposed to the sun, the UTCI
shows a stronger sub-grid spatial variability, in particular during the morning and afternoon, with the potential for stronger
heat stress than in the high-density neighbourhood. During night-time, the spatial variability is reduced, due to reduced MRT
variation as the shadowing effect disappears, and higher UTCI values are found at the high urban density location. This
difference in behaviour between the two locations can be seen also in Fig. 7, where the fractions of the 10th percentile of UTCI
values (i.e. representative of one of the coolest spots in the grid cell) and the 90th percentile (i.e., one of the hottest) in the
different heat stress regimes are shown for the two points. Here we can see that in the low-density urban point, the cool location
is in a comfortable UTCI range most of the time, while the hot (90th percentile UTCI) sub grid location is under stress most
of the time. On the other hand, less variability is present in the high-density neighbourhood, with fewer extreme values, and
most of the time in the strong or moderate heat stress regime for both the cool and hot locations within the grid square. This
kind of detail is not available from the Heat Index distribution which does not account for the mean radiant temperature, wind,
or their variabilities (Fig. 8).

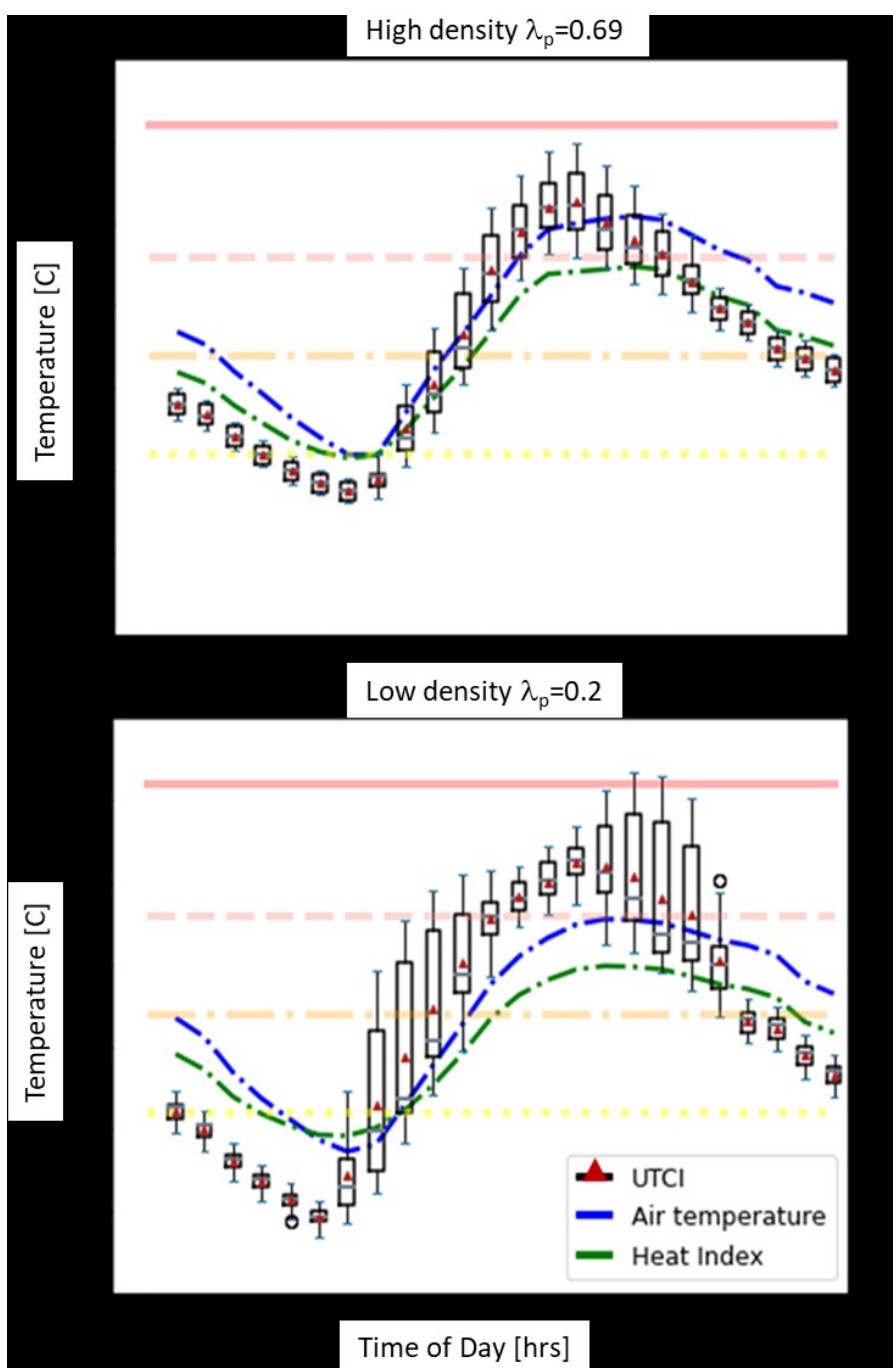

**Figure 6.** Diurnal evolution of UTCI compared with 2-m air temperature and Heat Index calculated from air temperature and relative humidity at each grid point). The UTCI boxplot at each hour represents the subgrid-scale distribution calculated based on 6 MRT, 3 wind speeds, and 3 air temperature values (54 combinations in total). The horizontal lines represent the thermal comfort zones for UTCI (i.e. above +46C: extreme heat stress; +38 to +46: very strong heat stress; +32 to +38: strong heat stress; +26 to +32: moderate heat stress; and +9 to +26: no thermal stress).

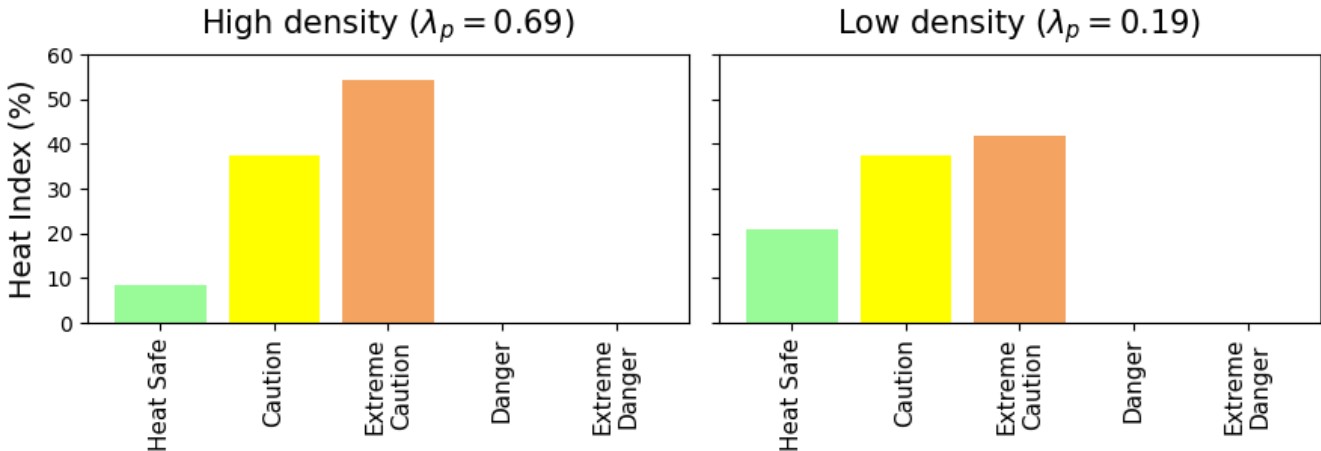



**Figure 7.** From top to bottom, the frequency of UTCI class over a 24-hour period, for a subgrid location that is cooler (i.e. 10th percentile
of UTCI in the urban canopy, top), and for a subgrid location that is hotter (i.e. 90th percentile of UTCI in the urban canopy, bottom), for
the high-density (left) and low-density (right) points.

**Figure 8.** same as Figure 7, but for the Heat Index

## 3.2 City-scale maps of outdoor thermal comfort and heat stress indicators.

The previous analysis helps to understand the spatial distribution of the different variables presented in Fig. 9 at 10 and 16 UTC (note that Madrid is at Longitude 3W, so UTC is essentially equal to solar time). In the dense city centre, the distribution of 2m air temperature at 0900 UTC shows a hot region, with cooler areas in the less dense regions around it. This effect is due to the fact that in the dense region, the reduced sky-view factor of the streets (high H/W), as well as the larger thermal storage capacity in the buildings, reduce the nocturnal cooling, and increase the vertical mixing in that part of the city compared to the surroundings. Such a difference is still visible in the morning. The higher temperatures in the S-E part of the urban area, and cool temperatures in the N-W are the result of the topographical differences. The spatial distribution of air temperature is qualitatively similar to the spatial distribution of the 10-percentile of UTCI (e. g. the cool spot in the grid cell), even if the differences between the centre and the surrounding urban areas are not as intense as for 2m air temperature. On the other hand, the 90-percentile map (hot spot), shows a completely different pattern; on the city centre, at that time of the day, the whole street is still in the shadow, while in the surrounding, less dense urban areas, there are points completely exposed to the sun. As a comparison, the map of surface temperature (a variable often used to represent the spatial distribution of heat in cities) as seen from a satellite, i.e. based only on a weighted average of roof, street, and vegetation temperatures (see full equations in Martilli et al. 2021), does not show a clear pattern, and it is uncorrelated with the other maps. This is a clear indication that this variable should not be used for the assessment of the heat hazard or heat stress in urban areas.

At 1600 UTC the air temperature shows again higher values in the city centre, lower in the urban surroundings, and a gradient from hotter S-E at lower elevations to cooler N-W at higher elevations (Fig. 10). Such a tendency is present also for the 10th percentile (cool spot), but with less variability. The 90th percentile map (hot spot) indicates that the area with elevated heat stress extends well beyond the city centre, including lower-density regions that, even if they have lower air temperatures, are fully exposed to the sun. Finally, as it was the case for 0900 UTC, the surface temperatures have a map uncorrelated with neither the air temperatures nor the UTCI maps.

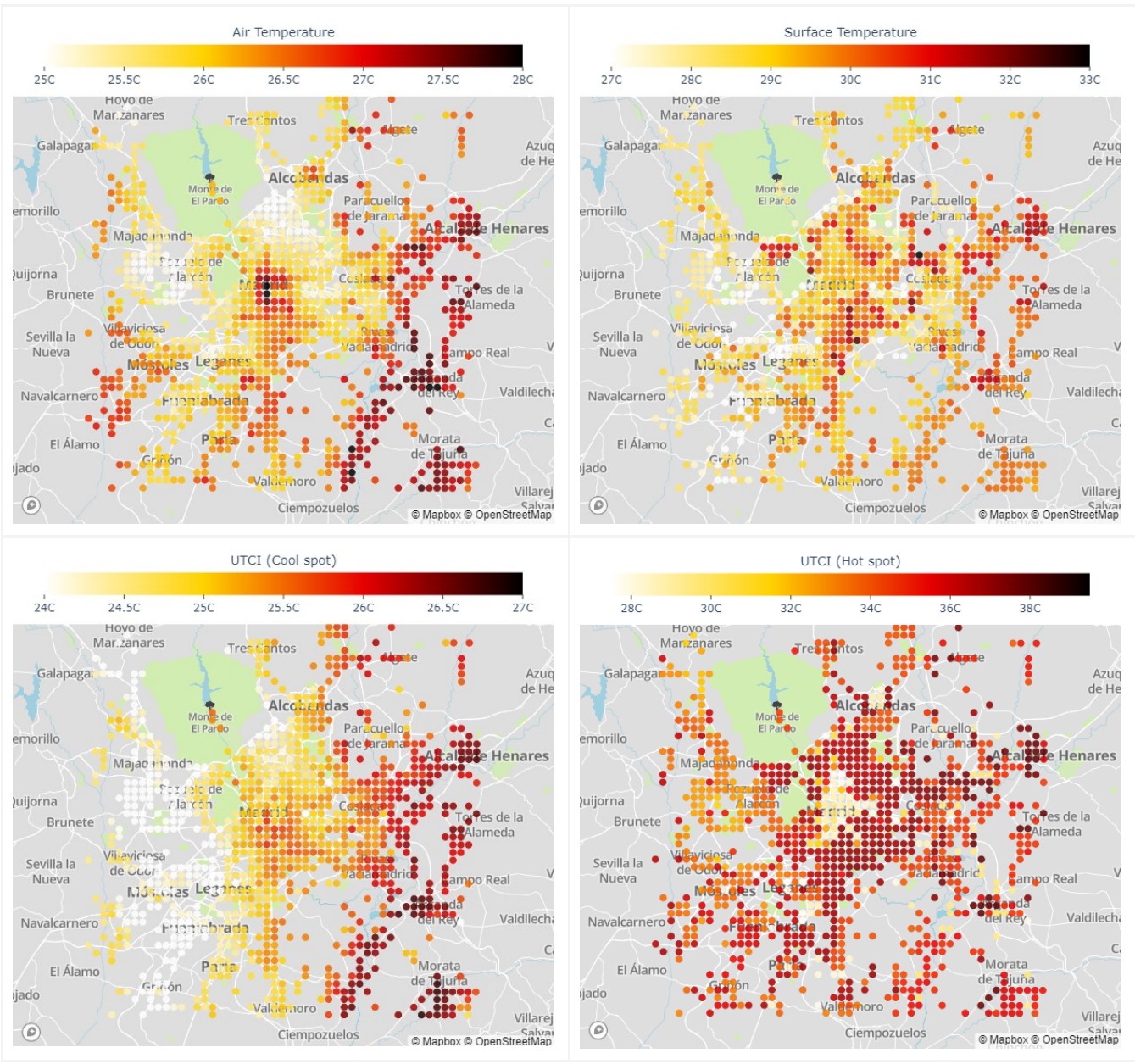

**Figure 9.** Spatial maps at 0900 UTC for 2-m air temperature (top left), surface temperature (top right), UTCI cool spot e. g. the 10 percentile of UTCI captured in the urban canopy model (bottom left), and UTCI hot spot e. g. 90 percentile of UTCI in the urban canopy (bottom right). Surface temperature is equivalent to that seen by a nadir-view satellite sensor (i.e., an area-weighted average of canopy ground temperature, roof temperature, and vegetation temperature in non-urban fractions is considered). The underlying maps were created with Mapbox OpenStreetMap



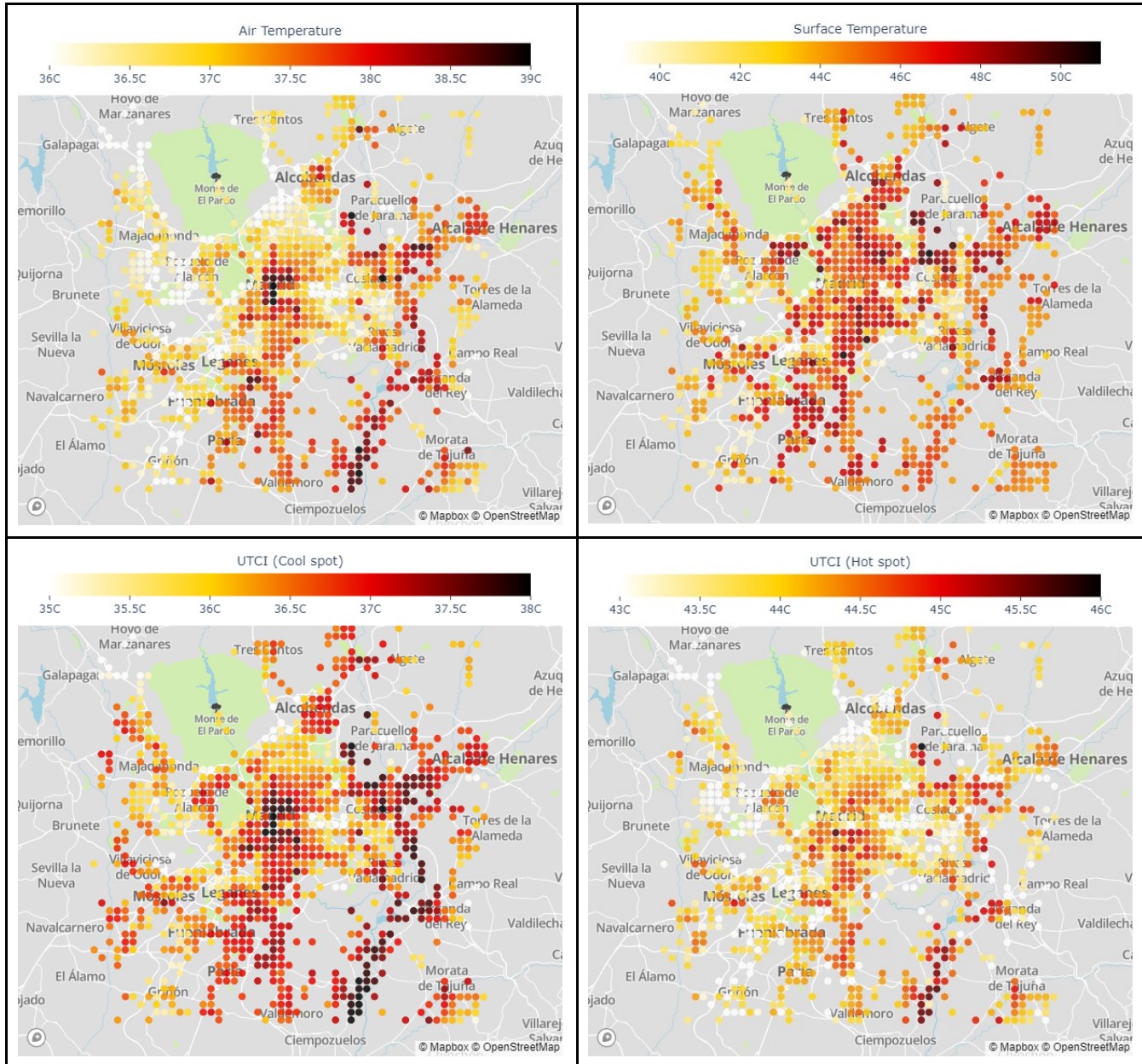



**Figure 10.** Same as Figure 9, but at 1600 UTC.

**4. Limitations**
The main limitation of the approach we propose here to account for the sub-grid variability of mean radiant temperature is the
idealization of the urban morphology adopted by the urban canopy parameterization BEP-BEM. This consists of representing
the urban morphology as a series of infinite urban canyons, all with the same width, separated by buildings of constant width,
and variable building height. Two street orientations are considered for each grid cell: North-South, and East-West. The
dimensions of the buildings and street canyons are determined such that the building plan area density, the density of urban
vertical surfaces per horizontal area, and the mean building height are equal to those of the real morphology of the grid cell.
As a result, the total surface areas of walls, roads and roofs in the idealized morphology used by BEP-BEM closely approximate
the corresponding surface areas in the real neighbourhood, and – to a certain extent – the street and buildings of the idealized
morphology can be considered representatives of an average street and set of buildings present in the grid cell. The advantage
of this approach, common among the most widely-used urban canopy parameterizations (Masson, 2000, Kusaka et al. 2001),
is that it allows accurate estimation of shadowing and radiation trapping effects in the urban canopy with low computational
cost, without considering the real urban morphology. Keeping the computational cost low was an essential requirement
considering the computational resources available when these urban canopy parametrizations were developed (about 20 years
ago). With today's computational resources, there may be potential to account for more complexity in the urban morphology.
However, this would require deep changes in the structure of the urban canopy parametrization BEP-BEM that are beyond the
scope of the present article. For this reason we decided to keep the idealized morphology of BEP-BEM and estimate the mean
radiant temperature in six locations representative of the middle of the street and the sidewalks. So, the mean radiant
temperatures computed are representatives of those six points of an "average" street in the grid cell. Indeed, in a grid cell of a
mesoscale model (that typically has a size of the order of one $km^2$) there is a variety of street and building dimensions and
orientations, so the present approach cannot capture the full spatial variability of mean radiant temperature, a variability that
increases with the heterogeneity of the real urban morphology. Nevertheless, it represents a step forward, since it accounts for
the range (and to some extent, the variability) of mean radiant temperature within the "average" idealized street canyon, that
can be reasonably considered the most likely street typology within the grid cell, something that previous approaches does not.
Overall, the current approach is likely to accurately quantify the mean radiant temperature of at least one "average" shaded
pedestrian and one "average" sunlit pedestrian (during periods with direct shortwave irradiance), and thus capture the largest
source of spatial variation of both MRT and UTCI (Middel and Krayenhoff, 2019). Another limitation of the approach
presented here is the lack of street trees. Currently work is in progress to introduce trees in the version of BEP-BEM
implemented in WRF via implementation of the BEP-Tree model (Krayenhoff et al. 2020), and in this way account for their
impacts on mean radiant temperature as well as on air temperature, humidity, and wind.
The approach used to estimate the mean wind speed and its sub-grid variability is grounded on a large number of CFD
simulations over a variety of urban morphologies. Indeed, as shown in Fig. 3, the sub-grid variability of wind speed can be
quite large, and certainly strongly influenced by the relative arrangements of buildings and streets. So, the approach presented
here will likely underestimate the sub-grid variability of wind speed – and this is why we decided to give the same likelihood
to the three values of wind speed estimated in (6), instead of assuming a Gaussian or Weibull distribution of the probabilities
of wind speed in the grid cell. To fully capture this variability a complete coupling between the mesoscale and a detailed CFD
model would be needed - something that we may be able to do in the near future, but is still unavailable with current
computational resources. Another limitation of the present approach is that the CFD simulations used to build the database
from which the parametrization has been derived are all for a neutral atmosphere, so thermal effects on wind speed and its sub-
grid variability are neglected.

## 5. Conclusions

A new parameterization to quantify intra-neighbourhood heat stress variability in urban areas using a mesoscale model is
presented. This approach is based on two primary developments: 1) calculation of mean radiant temperature at several locations
within the idealized urban morphology used by the urban canopy model BEP-BEM; and 2) parameterization of mean wind
speed and its sub-grid spatial variability as a function of the local urban morphology and the mean wind velocity computed by
the WRF mesoscale model, using relations developed from a large suite of CFD simulations over a range of realistic and
idealized urban neighbourhoods. The components of the new parameterization have been validated against microscale model
results. From this approach the sub-grid variability of a heat stress index (i.e. UTCI or SET) can be computed for every grid
point, permitting quantification of the heat exposure at both cool and hot locations within each grid square at each time.
The new parameterization has been implemented in the multilayer scheme BEP-BEM in WRF and used to simulate a heatwave
day over Madrid (Spain) as proof of concept. The results of this initial application demonstrate the following:
I. The new parameterization gives information that is more suitable for the evaluation of heat stress than the air
temperature, being based on an index (UTCI or SET) that also combines air humidity, wind speed, and mean radiant
temperature.
II. The new parameterization provides substantively more information than air temperature alone (or any other index
that does not account for the mean radiant temperature). It provides information about the sub-grid variability (such
that heat stress in both cool and hot locations in each grid square is quantified). To our knowledge, this has not been
done before with a mesoscale model.
III. The results for the investigated case indicate a strong intraurban variability, both in air temperature and UTCI values,
that can be linked to the differences in urban morphology and elevation above sea level. The ability to assess the
differential impacts of urban morphology on heat stress is key to the provision of guidance for urban planning
strategies that mitigate urban overheating.
IV. Nadir-view surface temperature (i.e., as seen from a satellite-mounted remote sensor) is poorly correlated with both
air temperature and UTCI maps, indicating that, despite its ubiquitous use at present, it is unlikely to be an adequate
metric for heat impact assessment studies.
Finally, we consider that this new development introduces a new methodology for deploying mesoscale models to assess urban
overheating mitigation strategies.


*Code Availability*

The code of WRF-comfort can be obtained here:

https://doi.org/10.5281/zenodo.7951433

The results of the simulation over Madrid shown in the manuscript are stored here:

https://zenodo.org/record/8199017

383

*Author contribution*

AM, NN, and ESK designed the methodology, AM developed the model code, AM and ARS performed the mesoscale simulations, JLach and ESK performed the TUF-3D simulations, JLu, ER, BS, JLS performed the microscale LES and RANS simulations, NN prepared the figures, AM, NN, and ESK prepared the manuscript with contributions from all co-authors.

*Competing interests*

The authors declare that they have no conflict of interest

390

391

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

## Appendix A. *Computation of Radiation for Mean Radiant Temperature*

As explained in the text, the mean radiant temperature at pedestrian level is represented using formula (1). The full expression of the longwave radiation components for the vertical faces of the pedestrian ($L_1, L_2$), for the case of an urban morphology with buildings of constant height and walls with no windows, is as follows:

$$L_1 = \sum_{i=1,n} \psi_{1i,p} \varepsilon_W \left( Rl_{1W_i} + \sigma T_{1i}^4 \right) + \psi_{1G,p} \varepsilon_G (Rl_G + \sigma T_G^4) + \psi_{1S,p} Rl_S$$

$$L_2 = \sum_{i=1,n} \psi_{2i,p} \varepsilon_W \left( Rl_{2W_i} + \sigma T_{2i}^4 \right) + \psi_{2G,p} \varepsilon_G (Rl_G + \sigma T_G^4) + \psi_{2S,p} Rl_S$$

Where (see Fig A1).:

$\psi_{1i,p}$ = is the view factor from wall section $i$ of building 1 to the side 1 of the pedestrian

$\varepsilon_W$ = is the emissivity of the wall

$Rl_{1W_i}$ = is the long wave radiation reaching the section $i$ of the wall of building 1

$T_{1i}$ = is the surface temperature of the section $i$ of the wall of building 1

$\psi_{1G,p}$ = is the view factor from the ground (or street) to the side 1 of the pedestrian

$\varepsilon_G$ = is the emissivity of the ground

$Rl_G$ = is the longwave radiation reaching the ground (street)

$T_G$ = is the surface temperature of the ground (street)

$\psi_{1S,p}$ = is the view factor from the sky to side 1 of the pedestrian

$Rl_S$ = longwave radiation from the sky

$\sigma$ = is the Stefan-Boltzmann constant.

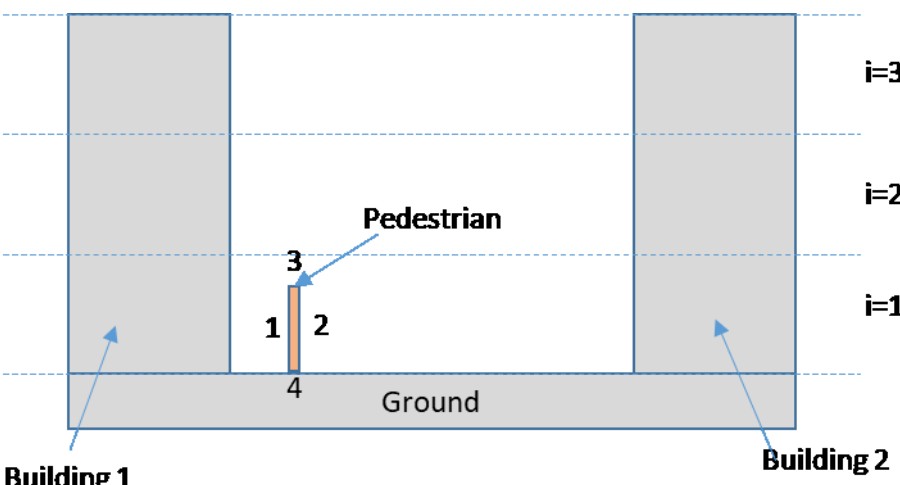

*Figure A1. Schematic of the Street canyon.*
Similar meaning applies for side and building 2.
The values of the surface temperatures and the longwave radiations are computed with BEP_BEM. The view factors are
estimated based on formulas A13-A19 of Martilli et al. 2002, using a height for the pedestrian of 1.8 m.
For the longwave radiation reaching the top of the pedestrian, we made the simple assumption that it is equal to the radiation
coming from the sky, $L_3 = Rl_S$, while for the longwave radiation reaching the bottom of the pedestrian, the assumption is that
it is equal to the radiation  emitted and reflected by the ground, or $L_4 = \varepsilon_G Rl_G + \varepsilon_G \sigma T_G^4$. We consider that these assumptions
are reasonable, giving that the contribution of the radiation reaching the top and bottom of the pedestrian is only 6% each to
the final value of the mean radiant temperature.
A similar approach is followed for the short wave radiation, leading to:

$$K_1 = \sum_{i=1,n} \psi_{1i,p} \alpha_i Rs_{1W_i} + \psi_{1G,p} \alpha_G Rs_G + Rs_{1S}$$

$$K_2 = \sum_{i=1,n} \psi_{2i,p} \alpha_i Rs_{2W_i} + \psi_{2G,p} \alpha_G Rs_G + Rs_{2S}$$
Where
$Rs_{1W_i}$=short wave radiation reaching the section $i$ of the wall of building 1
$\alpha_i$=albedo of the section $i$ of the wall of the building
$Rs_G$= is the short wave radiation reaching the ground
$\alpha_G$ = is the albedo of the ground
$Rs_{1S}$= is the short wave radiation from the sun reaching directly side 1 of the pedestrian, computed using formula A10 of
Martilli et al. 2002, using a height of the pedestrian of 1.8m.
Similar meaning for side and wall 2.
Regarding the radiation reaching the top of the pedestrian, $K_3$, for simplicity only the radiation coming directly from the sun
is considered, without accounting for the reflection from the walls. So the value is zero if the pedestrian is in full shadow, and
to estimate it, the formula used is from A11 of Martilli et al. 2002. The value of the radiation reaching the bottom of the
pedestrian is the value reflected by the ground, or $K_3 = \alpha_G Rs_G$,



Appendix B. *Comparison of Short wave calculation in BEP-BEM and TUF-pedestrian*.
Short wave radiation is an essential component of the MRT. Below we compare the short wave radiation reaching the vertical
sides of the segment representing the human body computed by BEP-BEM vs those estimated with the more detailed model
TUF-pedestrian.

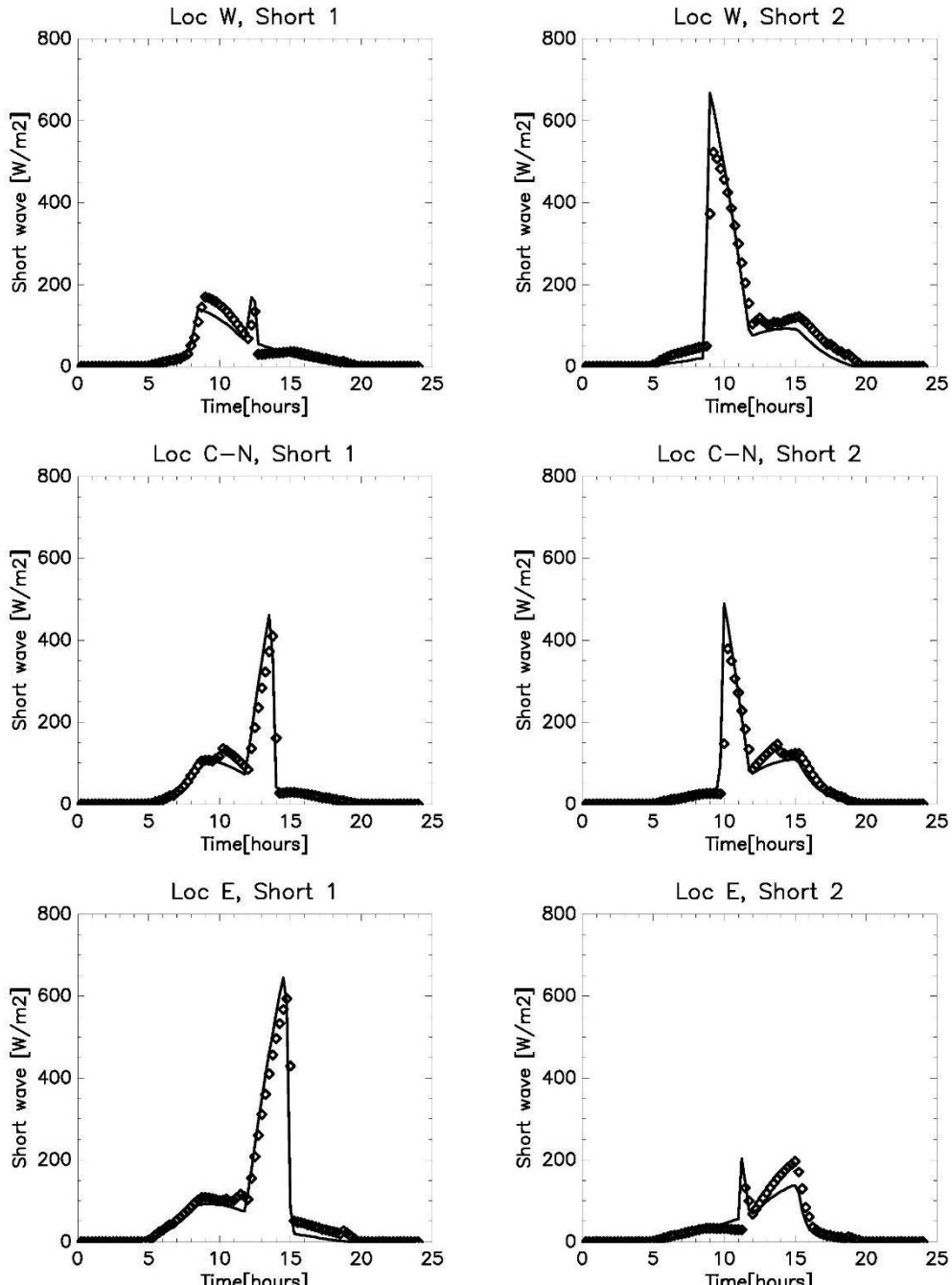

Figure B1. Comparison of short wave radiation at the two sides of the vertical segment representing the pedestrian for the N-
S oriented street. Solid line is the WRF, while diamonds are TUF. Short 1 means the side 1 of the pedestrian, while Short 2
the side 2.

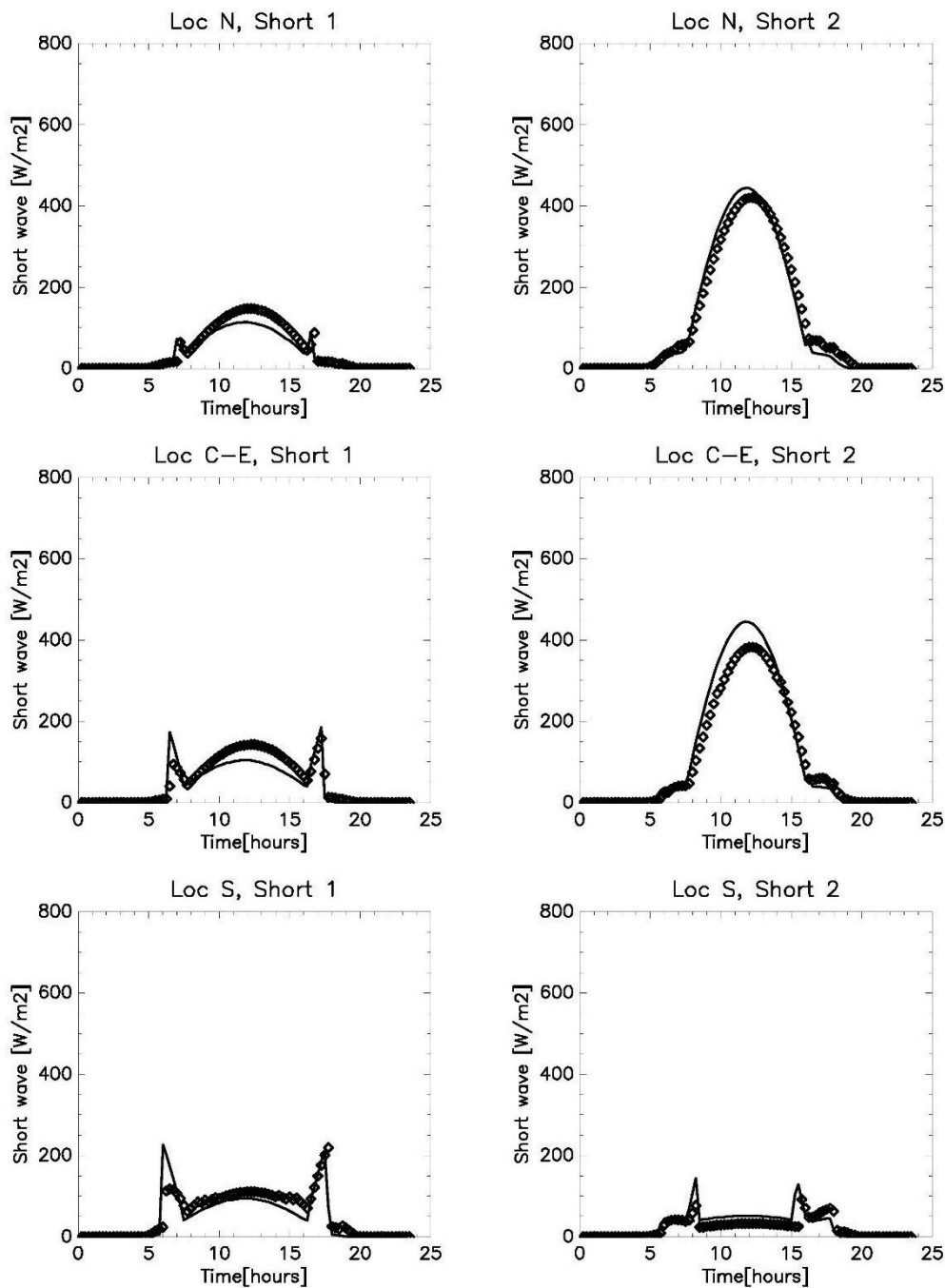



Figure B2. Same as B1, but for an E-W oriented street

Appendix C. *CFD simulations for wind speed variability*
Data from over 173 microscales CFD simulations of urban airflow are considered over realistic and idealized urban
configurations, spanning a wide range of building plan area ($\lambda_P$), frontal area ($\lambda_F$), and wall area ($\lambda_w$) densities representative
of realistic urban neighborhoods in different types of cities. CFD simulations are conducted using 162 large-eddy simulations
(LES) and 11 Reynolds-averaged Navier–Stokes (RANS) schemes detailed in Table B.1.

Table B.1 Details of CFD microscale simulation cases considered in this study. Simulations are classified based on the configuration (urban form) used. These classifications include **UA** (**U**niform height with **A**ligned configuration), **US** (**U**niform height with **S**taggered configuration), **VA** (**V**ariable height with **A**ligned configuration), **VS** (**V**ariable height with **S**taggered configuration), **UR** (**U**niform height with **R**ealistic configuration), and **VR-WD** (**V**ariable height with **R**ealistic configuration and multiple **W**ind **D**irections considered).

| Model | Classification | $H_m$[m] | $H_{max}$ [m] | $\lambda_p$ range | Count | Source | Example |
|---|---|---|---|---|---|---|---|
| LES | UA | 16 | 16 | [0.0625 - 0.64] | 7 | Nazarian et al. 2020<br>Lu et al. 2022 | |
| LES | US | 16 | 16 | [0.0625 - 0.64] | 7 | Nazarian et al. 2020<br>Lu et al. 2022 | |
| LES | VA | 16 | 20, 24 | [0.0625 - 0.64] | 42 | Lu et al. 2022<br>Lu et al. 2023 | |
| LES | VS | 16 | 20, 24 | [0.0625 - 0.64] | 42 | Lu et al. 2022<br>Lu et al. 2023 | |
| LES | UR | 16 | 16 | [0.057 - 0.536] | 64 | Lu et al. 2022 | |
| RANS | VR-WD | 14.5-34 | variable | [0.190 - 0.680] | 11 | Sanchez et al. (2017)<br>Santiago et al. (2017)<br>Kracht et al. (2017)<br>Borge et al. (2018)<br>Kracht et al. (2019)<br>Santiago et al. (2020)<br>Sanchez et al. (2021) | |


In the LES simulations, airflow over idealized and realistic urban arrays to determine the model parameters (Nazarian et al.,
2020; Lu et al., 2022, 2023).  Realistic urban layouts are prepared by rasterizing building footprints from an open-source
dataset OpenStreetMap using OSM2LES (Lu et al., 2022). 64 realistic urban neighbourhoods were obtained assuming uniform
building height (Table B.1) from several major cities such as Sydney and Melbourne (Australia), Barcelona (Spain), Detroit,

Los Angeles, and Chicago (United States). Idealized urban arrays are considered in aligned and staggered arrangement that follows (Coceal et al., 2007) with varying urban density ($\lambda_p$ in [0.0625,0.64]) and height variability ($H_{std}$=[0m,2.8m,5.6m]). Simulations are conducted in the Parallelized Large-eddy Simulation Model (PALM, version r4554) (Maronga et al., 2020) following the same setup in (Nazarian et al., 2020), which has validated results against Direct Numerical Simulation (Coceal et al., 2007) and wind tunnel experiments (Brown et al., 2001). The computational domain is discretized using the second-order central differences (Piacsek and Williams, 1970) where the horizontal grid spacing is uniform and the vertical spacing follows the staggered Arakawa C-grid. The minimal storage scheme is employed in the time integration to solve the filtered prognostic incompressible Boussinesq equations where the pressure perturbation was calculated in Poisson's equation and was solved by the FFTW scheme (Frigo and Johnson, 1998).

The RANS dataset is derived from steady-state CFD-RANS simulations performed with the Realizable k- ε turbulence model (STAR-CCM+, Siemens) over realistic urban areas. The size of the computational domains is determined following the best practice guideline of COST Action 732 (Franke et al., 2010). The horizontal area covers around 1-1.5 km2 and the domain top is at around 8H, being H the mean height of buildings. The resolution of the irregular polyhedral mesh used in all CFD-RANS simulations goes from 0.5 m close to buildings to 6 m out of the built-up area, which results in between 3 and 8 million grid points depending on the complexity of the geometry. Inlet vertical profiles for wind speed, turbulent kinetic energy (k), and its dissipation (ε), are established in neutral atmospheric conditions. The evaluation of the CFD-RANS simulations was addressed in previous studies summarized in Table B2 and more information is provided in previous publications.