# Peer review of "WRF-Comfort: Simulating micro-scale variability of outdoor heat stress at the city scale with a mesoscale model"

_EGUsphere, 2023_

## Author Comment (AC1)

RC1.

**General comments**

Quantifying outdoor heat stress of urban inhabitants requires consideration of air temperature and humidity, mean radiant temperature (MRT) and wind speed. As noted in the abstract, temperature and relative humidity vary on scales of hundreds of meters but wind speed and MRT vary at the meter scale. As a result, current simulation workflows require use of a mesoscale simulation to provide boundary conditions for time-consuming microscale computational fluid dynamics (CFD) simulations that, in the absence of supercomputer resources, apply only to a specific neighborhood for a short period of time. The authors argue for an alternative approach that relies on an estimation of MRT at several locations in streets of varying orientation, microscale CFD simulations to determine mean wind speed and its variation across a range of urban morphologies, and mesoscale simulation of temperature and relative humidity to estimate thermal-stress indicators and their variation at the sub-grid level of the mesoscale simulation.

While the method is more efficient than applying microscale simulations at large temporal and spatial scales, it is more complex than simple estimates of air temperature and humidity, which existing mesoscale simulations can efficiently provide over long time scales, and more demanding than analysis of remotely estimated surface temperatures. Here the authors argue that heat stress depends on more than air temperature and humidity and that surface temperatures estimated from satellite measurements do not correlate with heat maps generated with multiple input variables.

The proposed approach has merit in meeting the needs of urban planners and designers to assess heat stress at fine spatial scale and expansive time scales on the order of years. As developed by the authors, the workflow has the rigor necessary to produce useful results.

The backbone of the workflow is the BEP-BEM multilayer urban canopy parameterization, developed by one of the authors, that includes a building energy model and is incorporated into the widely used mesoscale Weather Research and Forecasting (WRF) model. The authors parameterize MRT at locations within the geometry and orientation allowed in BEP-BEM. The

parameterization of wind speed is based on a large number of CFD simulations (both Reynolds Averaged Navier-Stokes and Large Eddy Simulations) that span realistic and idealized urban configurations, with wind speed parameterized by the building roof area or, more satisfactorily, vertical wall area, each normalized by the total horizontal area in a grid cell.

From an assumed variation of 1 °C in air temperature and computed variations in wind speed and MRT, the authors calculated 54 combinations from which they calculated two measures of heat stress and documented the 10th, 50th and 90th percentile values.

The paper effectively communicates the application of the method for a typical heatwave day in Madrid, Spain, with diurnal plots of MRT, air temperature and UTCI and Heat Index for high- and low-density neighborhoods and bar charts of thermal stress categories as estimated by UTCI and Heat Index. The conclusion is a crisp summary of the method and results, establishing the contributions of the research. In all, the paper is coherent and effectively claims a significant addition to the modeling of urban outdoor stress.

*Authors: We thank the reviewer for their comments. Below are our answers to the specific comments*

Specific comments

1 )BEP-BEM is limited to two street orientations, each with the same street width and building height distribution. This limitation defines the considered variation in MRT, which is computed for three positions (sidewalks on opposite sides of the street and street center) for each of the two street orientations.  BEP view factors and shading algorithms are used to estimate shortwave reflection and longwave emission and reflection.  MRT accounts for shortwave and longwave radiation reaching a pedestrian, weighting radiation received from body surfaces at different orientations.  Not stated in the paper is the calculation of surface temperatures, which depends on a heat balance in which absorbed radiation can be emitted as longwave radiation or conducted into building material. Model results are validated by comparison with more detailed simulations made with the measurement-validated TUF-Pedestrian, but only over the designated locations in a street canyon.  Modestly more explanation of the physics in the reference model would have better bolstered the asserted confidence in the streamlined methodology developed in the paper.

*Authors: An appendix has been added with more details about how the different radiation components used to estimate the mean radiant temperature are computed.*

2) Variations over a wider range of geometries derived from the detailed simulation would have determined whether the range of MRT as constrained by the BEP geometry is a reasonable approximation.

*Authors: We agree with reviewer's comment, but we think that this limitation is intrinsically linked with the idealization done by the urban canopy paramterization (UCP) of the urban morphology. Giving that changing the idealized urban morphology used by the UCP is beyond the scope of the article, we consider that the approach presented is able to characterize – at least – the coolest and hottest spots in the grid cell, with reasonable accuracy. This point has been added in the **Limitations** section (4).*

3) Spatial maps at two times of the simulated day make a strong case that air and surface temperatures do not accurately predict UTCI values in hot-spot locations. The display of Heat Index is puzzling, because the paper promised the use of SET and UTCI but does not present simulated values of SET.

*Authors: The Heat Index was introduced as an example of an index that can be derived from standard mesoscale model outputs and accounts for moisture in addition to air temperature, but does not take into account MRT. We clarified this in the text.*

4) What's missing is a thoughtful discussion of limitations and, perhaps, a more detailed comparison with appropriate ground-truth simulations (with CFD and energy balances) for a single neighborhood. Do the authors think that the MRT model is adequate for all possible building materials and morphologies? What about the impact of trees?  Similarly, is the parameterization of wind speed universally applicable or would city planners need to conduct or commission local simulations?

*Authors: We thank the reviewer for this suggestion. A discussion of limitations has been added as Sect. 4.*

Editorial comments

Line 47. Please replace 1 with one.

*Authors: done.*

Line 65 uses "autonomy" to characterize inhabitant choice of thermal environment, a different use of the word than in "spatial autonomy," which refers to the extent, spatial or temporal" a space is thermally comfortable. People have agency to make choices, but spaces do not.

*Authors: Please review Nazarian et al. 2019 referenced here for detailed descriptions of the term outdoor thermal comfort autonomy (OTCA) and the spatial and temporal extent of that used in the literature.*

Line 71. Please hyphenate "grid average" (proper when followed by a noun).

*Authors: done*

Line 101. The sky is also a source of shortwave radiation, in which a portion of direct radiation from the Sun is scattered in the atmosphere.

*Authors: the source of shortwave radiation in the sky is the sun – in any case we decided to speak only of short and longwave radiation in the sentence that now reads:*

*For shortwave reflection and longwave emission and reflection, the standard BEP view factor and shading routines (Martilli et al. 2002) are used to estimate the amount of shortwave (direct and diffuse) and longwave radiation reaching a vertical segment 1.80 m tall and located in each of the six positions previously mentioned (Fig. 1, Appendix A).*

Line 119. In Equation 1, aK and aL are not defined.

*Authors: These are the absorptivity of the pedestrian in the shortwave and longwave, respectively. These definitions have been added in the text.*

The text asserts excellent agreement between two models for shortwave radiation loading but the figures in Appendix A show peak differences as much as 100 W/m2.

*Authors: considering the strong idealization of the urban morphology used by BEP-BEM we believe that these differences are acceptable. We change the sentence as follows:*

*"A comparison of the shortwave radiation loading on the pedestrian between the two models reveals very good agreement (Appendix B Fig. B1, B2), considering the highly simplified urban morphology used by BEP-BEM, with biggest errors limited to short periods of time"*

Line 147. Please replace "module" with "modulus."

*Authors: done*

Line 154. Please consider "Data are considered from over 173 microscale CFD simulations of urban airflow over realistic and idealized urban configurations,…"

*Authors: done*

Line 177. Please consider deleting  the commas that bracket "therefore."

*Authors: done*

Line 185. For consistency with other choices of tense, please replace "used" with "use."

*Authors: done*

Line 191. Please replace "Where" with "where."

*Authors: done*

Line 202. Please consider "that increasing cause severe heat stress…."

*Authors: done*

Line 203. Please separate "27,9,3" with appropriate spaces.

*Authors: done*

Line 228. Grid cell or grid point would appear to be better than grid, to describe a specific location.

*Authors: done*

Lines 231, 232, 244 and 247 use heat stress index, Heat Index, Heat index and heat index; please consider more consistency.

*Authors: changed to Heat Index*.

Line 263. The time here is stated as 9 UTC while in line 281 it is 09000 UTC.  For consistency with the afternoon time of 1600 UTC and the caption to Figure 9,  both should be 0900. Please consider starting the sentence with "In the dense region,…."

*Authors: done*

Line 270. Please consider "…completely different pattern; on the city center at that time of day,…."

*Authors: done*

Line 309. Please consider "…this has not been done before…."

*Authors: done*

Line 311. Please delete the comma after "case."

*Authors: done*

Line 454. The caption for Figure A1 should define Short 1 and Short 2. In the caption for Figure A2, please replace "a E-W…" with "an E-W…."

*Authors: done*

Reviewer 2:

This manuscript describes how to calculate outdoor heat stress across a city using the atmospheric mesoscale model, WRF. This manuscript deals with spatial variabilities in mean

radiant temperature (MRT) and wind speed in urban canyons in a decisive manner by simplifying their many aspects and assumptions. Originally such spatial variations cannot be resolved in the mesoscale model. The ideas proposed in this manuscript are interesting but simplified too much. Further investigation of this study needs important things below.

*Authors: We thank the reviewer for their comments. Below are our answers:*

1) The key idea of this study comes from the six-directional weighting method by (Thorsson et al., 2007). But this reference is missing in the manuscript and the description of this method is quite descriptive. This journal is for the code and model, and we expect more detailed information and code description.

*Authors: We added the references relevant to the six-directional weighting method. We also added an appendix with more details about the calculation of the radiation components of the Mean Radiant Temperature.*

2) The similar problem also goes to wind speed calculation in 2.2.

*Authors: The parameterization proposed in section 2.2 is based on a fitting of the data calculated from a large set of CFD simulations, as described in some detail in Sect. 2.2. The data of the CFD simulations are about to be made public, and are the subject of another paper entitled "UrbanTales: A comprehensive dataset of Urban Turbulent Airflow using systematic Large Eddy Simulations".*

Additionally, some information is vague. For example, what is the meaning of "close to the pedestrian height (~2.5 m)"? The symbol "~" stands for the approximation and why we need this approximation? So wind speed is at 2.5 m above the road?

*Authors: We use the symbol for "approximate" 2.5m above ground, because WRF uses sigma pressure terrain following coordinates, and therefore the thickness of the lowest model level slightly changes with the position (e. g. topography) and time. In the simulations presented the thickness of the lowest model level at sea level and standard pressure is 5m, and – since WRF uses an Arakawa C grid – the wind components are defined at 2.5m.*

1) What are the implications and limitations to use spatially averaged wind speed with estimation of MRT at three different locations? We need more considerate discussion on many assumptions and parameter values used in this study.

    *Authors: a discussion on the limitations has been added (Section 4).*

2) Figure 2 for the model evaluation may not be useful because the proposed model is based on the two-street orientation. We can also argue that parameters in the proposed model is calibrated in some sense to match the results. It will be quite useful if there is comparison between the model and in-situ data in a city.

    *Authors: The aim of the comparison shown in Fig. 2 is more verification (check if the model is working properly), rather than validation (check if the model represents the reality). This is why it is important that both BEP-BEM and TUF-3D*

*have the same urban morphology. The comparison shows that – for the same urban morphology – BEP-BEM gives very similar MRT values to TUF-3D. The limitations of the idealized morphology and the two street orientations are now discussed in section 4.*

*It is important to mention, also, that TUF-3D has been validated with measurements (Lachapelle et al 2022; Jiang et al. 2023) of MRT for neighborhoods in Phoenix, AZ and Guelph, ON. However, those measurements are at the microscale and cannot be used for validation of the BEP-BEM approach presented in the paper, which aims to assess the variability of MRT within the grid cell of a mesoscale model (order of one $km^2$). We are not aware of suitable experimental datasets to evaluate the MRT calculated by WRF-Comfort at the neighbourhood scale (i.e., spatial max and min MRT, for example).*

*In summary, Fig. 2 shows that 1) once the morphology is fixed, the computation of MRT done in BEP-BEM is correct, and 2) with all the limitations linked to the idealized morphology mentioned in section 4, we can expect reasonable values of MRT from BEP-BEM.*

3) Please check carefully if description on variables, abbreviation, and indices are well described in the manuscript.

*Authors: done*

---

## Referee Report (RR1)

EGUsphere 2023-1069 revision 1
WRF-Comfort: Simulating micro-scale variability of outdoor heat stress at the city scale with a mesoscale model

The authors have fully responded to each comment in my review of the initial submittal and made appropriate changes in an excellent manuscript.  Notably, the paper now includes an additional appendix to explain in detail the calculation of mean radiant temperature and a section in the text that identifies limitations of their method.  Well done!

---

## Referee Report (RR2)

**EGUsphere 2023-1069 final version**
**WRF-Comfort: Simulating micro-scale variability of outdoor heat stress at the city scale with a mesoscale model**

In revision 1, the authors fully responded to each comment in my review of the initial submittal and made appropriate changes in an excellent manuscript.  The final version includes a small number of changes that I also consider acceptable.

Editorial comments (concerning text added to the final version):

1.  Line 155.  Please replace "microscales" with microscale."
2.  Line 165.  Please remove the comma after "canopy" or match it with a comma after "fact."
3.  Line 167. Please remove the space after $\sigma_u$ and either remove the following comma or add one after "therefore."  Please consider replacing "that" with "which."
4.  Line 228.  Please consider "The map is oriented such that left is west and up is north; the size is 50x50 km."

---

## Author Response (AR2)

Review of the manuscript egusphere-2023-1069

WRF-Comfort: Simulating micro-scale variability of outdoor heat stress at the city scale with a mesoscale model

By Martilli et al.

Summary: In this study the authors use WRF model simulations using the BEP-BEM scheme at 1 km grid spacing for the city of Madrid to estimate human thermal comfort indices. This is done through estimating ranges for the mean radiant temperature, wind speed and temperature within a grid cell. This spans up 54 possible thermal comfort options for every grid cell and every time step. From this distribution the 10th, 50th and 90th percentile are presented to indicate mean human thermal comfort as well as for cool spots and for hot spots. I find this an interesting and new way to quantify the variability in human thermal comfort indices in a computationally cheap way, so in principle support publication of this manuscript. However, some aspects could be improved or advanced without much effort.

Recommendation: Minor revision required

Major remarks:

1. Obviously the modelling effort is a nice extension of what has been done before, but it lacks a verification against observations. It would be good to add some general model validation for the WRF simulation for the specific days, like performance for the airport + sounding at Barajas, as well as for the surface weather stations within Madrid.

   *Answer: WRF validation is not the objective of the paper. WRF with this set-up has been used and validated extensively for summer (Salamanca et al. 2012, Brousse et al. 2016), and winter periods (Martilli et al. 2021). The simulations used in this paper, have been validated in Rodriguez-Sanchez (2020), with 5 meteorological stations located in the urban area of Madrid. Below are the RMSE and BIAS for these stations:*

| station | BIAS (Celsius) | RMSE (Celsius) |
|---|---|---|
| Centro Municipal Acústica | 4.02 | 4.24 |
| Junta Municipal de Distrito Hortaleza | 0.58 | 1.06 |
| Estación Depuradora de Aguas Residuales La China | -0.90 | 1.12 |
| Junta Municipal de Distrito Moratalaz | 1.44 | 1.84 |
| Junta Municipal de Distrito Villaverde | 0.98 | 1.74 |

*With the exception of station 1, located close to the Manzanarre's river not resolved by the model, the statistical error for the other stations is in the range of what is commonly found for mesoscale models.*

*Ref:*
*Brousse, O., Martilli, A., Foley, M., Mills, G., & Bechtel, B. (2016). WUDAPT, an efficient land use producing data tool for mesoscale models? Integration of urban LCZ in WRF over Madrid. Urban Climate, 17, 116-134.*
*Martilli, A., Sanchez, B., Rasilla, D., Pappaccogli, G., Allende, F., Martin, F., ... & Fernandez, F. (2021). Simulating the meteorology during persistent Wintertime Thermal Inversions over urban areas. The case of Madrid. Atmospheric Research, 263, 105789.*
*Salamanca, F., Martilli, A., & Yagüe, C. (2012). A numerical study of the Urban Heat Island over Madrid during the DESIREX (2008) campaign with WRF and an evaluation of simple mitigation strategies. International Journal of Climatology, 32(15), 2372-2386.*

2. I find the discussion section can be strengthened by discussing whether less than 54 combinations of meteo input for the UTCI calculations would also do the job, or would let's say 108 do a better job? Or would 27 work as well? In other words: one need to discuss how robust are the estimated subgrid-scale distributions of UTCI.

*Answer: 54 is coming from the combination of 3 wind speeds, 3 air temperatures and 6 mean radiant temperatures. The calculation is done only at the time when the output is printed, and so is not a big penalty in terms of CPU. We consider this as the minimum amount of values to account for the variability in the grid cell. In particular, for the mean radiant temperature, the magnitude with greater spatial variability, we consider, for each street direction, the two locations close to the walls (the most likely to be either on full shade or full sun, and therefore representing the extreme values, and also a location where people usually are), plus the value in the middle of the street. In order to check this, an idealized simulation has been done where the mean radiant temperature is computed in 5 points, instead of 3, for each street direction. The new points added are half way between the one in the center of the street and those close to the walls (for a 14m wide street, as the one used in the test, this means that the points are at 1.5m, 4.25m and 7m from each wall). The statistics is therefore computed over 90 values (3X3X10) instead of 54. The results obtained are shown in the graphs below. They represent the time evolution over 24hrs of the 10$^{th}$ percentile (first plot), the 50$^{th}$ percentile (second plot), and 90 percentile (third plot) of the simulation with 5 mean radiant temperatures per street direction (e.g. 90 values of UTCI) in black, and the original version with 3 mean radiant temperatures (54 values of UTCI) in red.*

FERRET (optimized) Ver 7.6
NOAA/PMEL TMAP
22-MAR-2024 08:33:32

X : 10
Y : 10

[Figure]

COMF_10[D=d_tmr]          COMF_10[D=d]

FERRET (optimized) Ver 7.6
NOAA/PMEL TMAP
22-MAR-2024 08:39:50

X : 10
Y : 10

[Figure]

COMF_50[D=d_tmr]          COMF_50[D=d]

[Figure]

*As it can be seen, the differences are small, indicating that adding more values to compute the statistics is not worth.*

Minor Remarks:

General remark: the authors call the method a "parameterization". One could discuss this is indeed a parameterization. Classically one uses the term parameterization to estimate a higher order moment from the lower order moment available on the grid. Would the term "downscaling" not fit better here?

*Answer: Here we consider "parametrization" an approach that allows to estimate processes that are too small scale to be resolved explicitly in the model. But the reviewer makes a good point, since, to a certain extent, our procedure "downscales" wind and mean radiant temperatures, so we re-phrased it at line 22 to reflect this.*

Ln 119: short -> short- (or shortwave)

*Answer: modified*

Figure 2: TUF-Pedestrian: perhaps add in the caption TUF-Pedestrian acts here as a reference.

*Answer: modified*

Figure 3: I have doubt about the extrapolation of sigma_u/U to be zero at vanishing lambda's. Classical boundary layer scaling for neutral flows says this ratio goes to a constant value, so for lambda_p =0.

*Answer: This is a very important point, and we thank the reviewer for asking this. The variability we refer here is for the **mean** value of the wind speed, where **mean** should be*

*intended as ensemble average (average over many realizations), or time average over time scales much longer than the turbulent time scales (but shorter than the time scale at which mesoscale features vary). Note that since there is spatial hetereogenity in the urban canopy, the classical Taylor hypothesis that space, time and ensemble averages are equal does not hold anymore. Space average is not equal to time or ensemble averages. However, lambda going to zero, means that there are no buildings, and so the space becomes horizontally homogeneous, and the **mean** wind must be the same in all the points of the space, implying that the sigma_u (standard deviation of the spatial variability of the mean wind speed) must be zero. The sigma mentioned by the reviewer is connected to the variability of the instantenous wind speed, and is induced by the turbulence, and indeed, is not zero for homogenous surfaces. We decided to neglect the impact of the turbulence since we make the assumption that the mean wind speed is the relevant quantity for thermal comfort – extending this method to the impact of the turbulence is left for future studies. This has been explained in the text (lines 161-168).*

Ln 178: please add a justification for limit/clipping introduced in speed1

*Answer: this is to avoid negative values for the wind speed.*

Ln 182: a simple log law… Please add a justification to use this. One cannot extrapolate the wind speed from within the canyon to the 10-m level using a simple log law.

*Answer. Clearly, the relevant wind for thermal comfort must be at the pedestrian level (e. g. around 2m), and not at 10m above ground. The reason why 10m is used in UTCI must be because this is the reference height in WMO standard measurements, which are the type of measurements usually available. The assumption we make is that the location where the UTCI formulation have been derived and tested is close to a WMO station, on open ground, where the log law is valid. Therefore, the relation between the relevant wind for thermal comfort (U2m), and the wind speed at 10m (U10m) at the location where UTCI has been derived and tested is $U2m = U10m \ log(2m/z0)/log(10m/z0)$ (neglecting the atmospheric stability). What we do in our approach is to inverse the formula and extrapolate U2m_mod (the wind computed by the model at 2m), to 10m above ground using the log formula, or $U10m\_ext = U2m\_mod \ log(10m/z0)/log(2m/z0)$. In other words, U10m_ext is the wind that – interpolated logarithmically – gives at 2m the wind speed produced by the model at that height (U2m_mod).*

Equation 1: in fact there is not justification for using 1 K temp variability. In classical boundary-layer theory sigma_T scales with theta_star, which depends on the sensible heat flux from the grid cell and the friction velocity in the grid cell, which are both available. So I think a physically more consistent temperature variance can be taken than was done here.

*Answer: Similarl, to what mentioned for the wind, we must keep in mind here that the sigma_T is not the variability of the instantaneous temperature induced by turbulence fluctuations, but the spatial variability of the **mean** value of temperature. Unfortunately, we do not have a complete set of non-neutral CFD simulations to assess this variability as we did for the wind speed. This should span not only different types of urban morphology, but also different atmospheric stabilities and solar angles (see for example Santiago et al. https://doi.org/10.1016/j.uclim.2014.07.008, Nazarian et al. https://doi.org/10.1007/s10546-017-0311-9) . We start to have also simulations over urban morphologies that resemble real ones (that were not available when the paper has been submitted almost one year ago), like the one performed by Esther Rivas (CIEMAT, personal communication) over a regular neighborhood of Madrid (barrio Salamanca) during a heat wave. These results indicate that for*

*this morphology the spatial variability of air temperature ranges between 0 C during the night, and 1.2 C during the day, which are in the same range of the 1C that we used as estimate. Indeed, here there is a huge possibility of improvement in the future, when these type of non-neutral simulations will become increasingly available. This has been mentioned in lines 194-197.*

Figures 4, 9, 10: please add scale bar and north arrow

*Answer: The map is oriented so that left is West, and up is North, and its size is 50x50km. This has been added in the caption.*

Ln 331: Gaussian distribution -> wind never follows a Gaussian distribution. So it is better to discuss here whether you could have better drawn the wind values from a Weibull distribution (the standard one for wind speed).

*Answer: Instantaneous wind speed follows a Weibull distribution, but here we are talking about a distribution of the mean wind speed. In fact, we do not know what kind of shape the distribution of the mean wind speed would follow in an urban canopy – probably it would be strongly sensitive to details of the urban morphology that are not captured by the urban canopy parameterization. This is why we decided to give the same probability to the three values.*

---

## Author Response (AR3)

Answers to referee's comment.

We thank the reviewer for her/his comments. All the editorial issues have been fixed.